# OpenReview forum: "RobustIT: Adapter-Centric and Attack-Agnostic Anti-Backdoor Instruction Tuning"
_ICLR.cc/2026/Conference — ICLR 2026 Conference Withdrawn Submission_

### Official Review · Reviewer_ckUZ · 2025-10-17

**Soundness:** 1
**Presentation:** 2
**Contribution:** 2
**Rating:** 4
**Confidence:** 4

**Summary:**

The paper presents a method to defend against data poisoning attacks. In this case the model is trained or fine-tuned using poisoned data, opening backdoor to attack the model during deployment. The proposed method consists of two regularizations, namely Input Diversity Regularization and Anomalous Activation Regularization. The method focuses on testing the defence in cases when fine-tuning of the model is performed via adapters.

**Strengths:**

•	The method is effective and it drives the attack success rates to essentially 0. At the same time it also boosts performance, presumably because parts of it could be seen as data augmentation and useful regularization to boost the learning ability.

•	The method is lightweight as the training times do not become significantly longer.

•	There are relatively extensive additional analyses.

•	The method does not need access to the attack details to work.

**Weaknesses:**

•	While the paper focuses on adapter-centric setups where adapters are inserted into the model and fine-tuned, it seems to me the method itself is more general and does not depend on this specific setup (e.g. AAR likely can apply to standard weights too). As a result, the focus on adapters feels less well-justified. Would it be possible to evaluate the method also in cases where fine-tuning without adapters is performed? Also only using IDR on its own gives almost all of the benefits (similarly also AAR, but slightly worse).

•	There are various existing defences (mentioned in the related work section), which presumably work quite well but are not compared against, because they do not focus on cases where only adapters are fine-tuned. Perhaps extending them to the adapter-centric case is easy and so they should be compared against? Using adapters quite often means we basically assume full access to the model, so adapter-centricity may not be a very crucial aspect.

•	The main part of the paper only evaluated one model, while it seems the experiments are not computationally expensive, only taking e.g. around half an hour or less if I understand correctly. It would be good to report the key results on multiple models. There are a few results in the appendix though.

•	The paper would benefit from proof-reading as the number of typos and other issues is relatively higher. E.g. in the abstract “fail in real-world, efficient tuning applications”, and in other parts “Falmingo”, “Updation”.

•	In general it feels the referenced work is less up to date, e.g. Flamingo is now one of the older MLLMs, while the paper states it is modern.

**Questions:**

•	Can the method be easily applied also for the case when full model is being fine-tuned or trained?

•	Is it easy to modify the existing defences for data poisoning into the setup where adapters are inserted?

•	What kind of adapters are considered? E.g. LoRAs are very common but it seems a different kind of adapter was considered. It could be useful to experiment with different types of adapters and show the method works there.

---

### Official Review · Reviewer_pTJS · 2025-10-27

**Soundness:** 2
**Presentation:** 2
**Contribution:** 2
**Rating:** 2
**Confidence:** 4

**Summary:**

This paper presents RobustIT, a defense framework designed to mitigate backdoor attacks during instruction tuning of vision–language models (VLMs). The method introduces two core components: Input Diversity Regularization (IDR), which applies data augmentation to increase input diversity and weaken trigger dependencies, and Anomalous Activation Regularization (AAR), which penalizes neurons that respond abnormally to backdoored inputs. The authors claim that RobustIT is attack-agnostic and adapter-centric, and evaluate it across seven backdoor attacks and multiple datasets.

**Strengths:**

- The proposed method is conceptually simple and easy to understand

- The topic of backdoor defense in vision–language models is highly relevant and important, given the growing deployment of multimodal systems.

**Weaknesses:**

- **Overclaim between the claimed scope and the actual experiments:**
There is a clear mismatch between the claimed scope and the actual experiments. The paper repeatedly positions its contribution as targeting Large Vision–Language Models (LVLMs) such as Flamingo, Otter, BLIP-2, and MiniGPT-4, but the experiments use outdated adapter-based models (e.g., frozen CLIP ViT-L/14 visual encoder). Most current LVLMs do not rely on Clip adapters. This inconsistency raises concerns about overclaiming in the **title, abstract, and introduction**. The authors should clarify the intended model scope and justify the use of these models.

- **Unclear justification for Input Diversity Regularization (IDR):**
The rationale behind using data augmentation to suppress triggers is not convincingly explained. The paper lacks both theoretical grounding and empirical verification of how specific augmentations mitigate backdoor activation. Furthermore, the robustness of the data augmentation based defense against stronger or adaptive triggers remains untested.

- **Limited validity of Anomalous Activation Regularization (AAR):**
   - The proposed assumption—that abnormal activations reliably capture trigger behavior—appears underexplored.

  - The setup (attack type, poisoning rate, affected layers) is insufficiently detailed, leaving ambiguity about the analysis conditions.

  - No cross-attack or cross-model comparisons are shown to verify generality.

  - If the attacker constructs semantically aligned backdoors (where poisoned and clean samples share similar semantics), the activation differences may disappear, and AAR could fail completely.

  - The authors should verify whether the observed activation patterns remain consistent across datasets and model architectures.

- **Incomplete baselines and comparisons:**

   - Table 1 lacks results without trigger inputs, which are essential to establish a fair ASR baseline, since large models occasionally exhibit unsafe generations even without triggers.

   - Figure 3 should use ASR (Attack Success Rate) directly instead of “100 – ASR,” which is non-standard and confusing.

   - In Table 4, the observation that β = 0 achieves the best defense and that disabling dynamic sparsification yields the lowest ASR seems counterintuitive and warrants explanation.

- **Insufficient coverage of poisoning rates and trigger scales:**
The evaluation is conducted only under a 1% poisoning ratio, which limits the reliability of the conclusions. Results under different poisoning rates (e.g., 0.5%, 2%, 5%) and different trigger sizes or intensities should be included to assess scalability and robustness.

- **Reproducibility concerns:**
The authors rely on several prior attack implementations but do not provide any released or referenced code for these baselines. Given the empirical nature of this work, reproducibility is crucial for validating both attack and defense effectiveness. Without access to baseline configurations, the experimental credibility remains uncertain.

**Questions:**

Overall, I find this work to have multiple unresolved issues, including the soundness of its **defense mechanism, the reasonableness of its assumptions, and the reproducibility of its experimental results**. I strongly recommend that the authors address these issues thoroughly. I will reconsider my rating after reading the authors’ response.

---

### Official Review · Reviewer_R9DC · 2025-11-01

**Soundness:** 3
**Presentation:** 2
**Contribution:** 3
**Rating:** 4
**Confidence:** 4

**Summary:**

This paper proposes RobustIT, a novel approach for anti-backdoor fine-tuning. RobustIT employs two regularization techniques, Input Diversity Regularization (IDR) and Anomalous Activation Regularization (AAR), to avoid the strong correlation between the backdoor trigger and the target response. Experiments demonstrate that both IDR and AAR are effective against multiple multi-modal backdoor attacks, with IDR + AAR performing the best.

**Strengths:**

1. This paper is easy to follow.
2. The methodology is simple yet effective.
3. Ablation on key components (IDR and AAR) and hyperparameters (adapter sparsity $\gamma$ and momentum $\beta$) is well presented.

**Weaknesses:**

1. The Equations are a little messy. Some symbols are not well-defined, e.g., $H$ in Eq. (3) and $f$ in Eq. (7). These symbols should be clearly defined when they first appear.
2. There is no experiment on different backbone models, weakening the discussion on the generalization of RobustIT. Currently, all experiments adopt the same backbone Otter-MPT1B-RPJama-Init, which may introduce bias when assessing RobustIT.
3. There is no clear definition of zero-shot and one-shot settings in experiments.
4. There is a lack of two critical ablation studies on the sample size and the poisoning rate (based on **the attacker's capability**). A key feature of RobustIT is attack-agnostic. Thus, both the sample size and the poisoning rate are uncontrollable for the defender. RobustIT adopts a 1% default poisoning rate and does not mention the sample size in the main paper. Although RobustIT performs well when the attacker poisons 1% of the total samples, it is not straightforward to infer that RobustIT will still perform well when the attacker poisons a significantly larger number of samples or when the attacker utilizes more samples to do fine-tuning. We need to carefully examine the boundary of RobustIT's effectiveness. Moreover, a poisoning rate of 0% serves as a good baseline for RobustIT to assess how IDR and AAR impact fine-tuning when there is no attacker present at all.

**Questions:**

Please refer to Weaknesses.

---

### Note · Authors · 2025-11-12

I have read and agree with the venue's withdrawal policy on behalf of myself and my co-authors.